# Crop Evolution of Foxtail Millet

**DOI:** 10.3390/plants13020218

**Published:** 2024-01-12

**Authors:** Kenji Fukunaga, Makoto Kawase

**Affiliations:** 1Faculty of Life and Environmental Sciences, Prefectural University of Hiroshima, Shobara 727-0023, Japan; 2Faculty of Agriculture, Tokyo University of Agriculture, Atsugi 243-0034, Japan

**Keywords:** center of diversity, crop evolution, domestication, genetic differentiation, phylogeny

## Abstract

Studies on the domestication, genetic differentiation, and crop evolution of foxtail millet are reviewed in this paper. Several genetic studies were carried out to elucidate the genetic relationships among foxtail millet accessions originating mainly from Eurasia based on intraspecific hybrid pollen semi-sterility, isozymes, DNA markers, and single-nucleotide polymorphisms. Most studies suggest that China is the center of diversity of foxtail millet, and landraces were categorized into geographical groups. These results indicate that this millet was domesticated in China and spread over Eurasia, but independent origin in other regions cannot be ruled out. Furthermore, the evolution of genes was reviewed (i.e., the *Waxy* gene conferring amylose content in the endosperm, the *Si7PPO* gene controlling polyphenol oxidase, the *HD1* and *SiPRR37* genes controlling heading time, the *Sh1* and *SvLes1* genes involved in grain shattering, and the *C* gene controlling leaf sheath pigmentation), and the variation and distribution of these genes suggested complex patterns of evolution under human and/or natural selection.

## 1. Hypotheses of the Origin of Foxtail Millet and Recent Advances in Foxtail Millet Genomics

Foxtail millet, *Setaria italica* (L.) P. Beauv., is one of the oldest domesticated cereals in the Old World. Recent archeological studies have indicated that foxtail millet originated in China [1]. Foxtail millet has been utilized in various ways, some of which are particular to respective areas of Eurasia [2], and it is thought to have played an important role in early agriculture in the Old World [3].

The geographical origin of foxtail millet remains controversial. Cytological studies have suggested that the wild ancestor of foxtail millet is the green foxtail (*S. italica* ssp. *viridis*, syn. *S. viridis*) [4,5]; however, the geographical origin of domesticated foxtail millet cannot be determined from the distribution of ssp. *viridis*, as this taxon is commonly found in various areas of Europe and Asia (and currently also in the New World). Vavilov [6] stated that East Asia, including China and Japan, is the principal center of diversity of foxtail millet. Harlan [7] proposed independent domestication in China and in Europe based on archeological evidence. Archeological, isozyme, and morphological evidence [8,9,10,11] suggest that China is the center of diversity and the place of presumed origin of foxtail millet, but independent origins in other regions cannot be excluded. Further, Li et al. [11] stated that landraces in Afghanistan and Lebanon may have been domesticated independently in relatively recent times because these landraces exhibited primitive morphological characteristics such as several tillers with small panicles and resembled ssp. *viridis* but had non-shattering large grains. Despite the primitive plant shape, a long cultivation history of Afghan landraces may be implied by their grain size, which is as large as that of some Chinese and Korean landraces [12]. Molecular analyses support the view that China is the center of foxtail millet diversity and that local landrace groups have differentiated after domestication (isozymes: Jusuf and Pernes [9]; prolamin [13]; rDNA [14,15,16,17]; RAPD [18,19]; AFLP [20]; genomic restriction fragment length polymorphism [RFLP] [21]; mitochondrial DNA RFLP [22]; transposon display [23]). Recent archeological evidence also supports the domestication of foxtail millet in China [1,24]. In contrast to the hypothesis of a Chinese origin and multiple origins, Kawase and Sakamoto [25] suggested that foxtail millet originated within the area ranging from Afghanistan to India because genetically less specialized accessions and those showing primitive morphological traits were found there. This hypothesis, which precludes China as the origin of foxtail millet, is a stark contrast to other theories.

The foxtail millet genome was sequenced by two research groups [26,27], and the genome sequence of its wild ancestor, ssp. *viridis*, was also published [28]. The diversity and evolution of foxtail millet have been investigated using whole-genome sequence information, and individual genes involved in foxtail millet domestication and diversification have been identified.

Here, we review studies on the genetic differentiation of foxtail millet landraces collected from various parts of Europe, Asia, and Africa in terms of morphological/agronomic characteristics, biochemical markers, intraspecific hybrid pollen sterility, and DNA markers, and we discuss recent studies on genes involved in the domestication and diversification of foxtail millet, focusing on *Waxy* conferring amylose content in the endosperm, *Si7PPO* controlling polyphenol oxidase, *HD1* and *SiPRR37* controlling heading time, *Sh1* gene and *SvLes1* which are involved in grain shattering, and the *C* gene controlling leaf sheath pigmentation.

## 2. Variation in Morphological Characteristics

Foxtail millet has diverse morphological and agronomic traits (Figure 1). Variations in the morphological and agronomic characteristics involved in the domestication and diversification of foxtail millet have been described and analyzed [8,10,11,12,29,30,31,32,33,34,35,36,37]. Some researchers have classified foxtail millet landraces into two to four subspecies, varieties, or races, such as *moharia*, *maxima*, *indica*, and *nana* [8,11,29,33]; however, the criteria for classification are ambiguous. Kawase [30] and Ochiai et al. [34] investigated variations in the morphological and agronomic characteristics of foxtail millet landraces from Europe and Asia, such as plant height, number of tillers, panicle length, and number of days to heading. Nguyen and Pernes [38] and Li et al. [11] also investigated variations in the morphological and agronomic characteristics of foxtail millet landraces using multivariate analyses. Their results indicated that morphologically primitive landraces are characterized by several tillers with small panicles, which resemble ssp. *viridis* but have non-shattering large grains, and are distributed in Afghanistan, northwestern Pakistan, Central Asia, and Lebanon, whereas most accessions from other regions, such as East Asia, have few or no tillers with one or more large panicles. Hammer and Khoshbakht [37] reported the cultivation of morphologically primitive landraces in Northern Iran. A few researchers claim that foxtail millet was domesticated in the region of Central Asia–Pakistan–Afghanistan–Northwest India because the morphologically primitive type was cultivated there [2,35], whereas Li et al. [11] were of the opinion that the morphologically primitive type with several tillers and small panicles, which resembles ssp. *viridis*, was domesticated independently. Description and analyses of morphological variation are important, but they are insufficient for addressing questions on the geographical origins and phylogeny of foxtail millet landraces.

## 3. Genetic Differentiation of Foxtail Millet Landraces According to Biochemical and Genetic Markers and Intraspecific Hybrid Pollen Sterility

Several studies have been conducted to clarify the genetic relationships of foxtail millet from Europe, Asia, and Africa based on (1) biochemical markers (isozymes and prolamin), (2) intraspecific hybrid pollen sterility, and (3) DNA markers (nuclear RFLP, mitochondrial RFLP, random amplified polymorphic DNA [RAPD], amplified fragment length polymorphism [AFLP], and transposon display markers [TD]), and single-nucleotide polymorphisms (SNPs). As summarized in Table 1, these studies revealed that foxtail millet landraces differentiated into local geographical groups, such as East Asia, Nansei Islands (Japan)–Taiwan–the Philippines, South Asia, and Europe, and that East Asian landraces (in particular, Chinese landraces) were the most diverse. 

Hunt et al. [40] determined the population genomic structure and relationship between domesticated lineages and green foxtail using genotyping-by-sequencing (GBS). Foxtail millet landrace accessions (*n* = 328) and green foxtail accessions (*n* = 12) were sequenced using GBS. They extended the geographic coverage of green foxtail by including previously published GBS sequence tags, yielding a 4515-SNP dataset for phylogenetic reconstruction. All foxtail millet samples were monophyletic relative to green foxtail millet, suggesting a single origin of foxtail millet, although none of the groups were clearly the most ancestral. Four genetic clusters were found within foxtail millet, each with a distinctive geographical distribution. Together with archaeobotanical evidence, these results suggest plausible routes for the spread of foxtail millet. 

Most studies suggest that China is the center of diversity of foxtail millet, and landraces were categorized into geographical groups, indicating that this millet was domesticated in China and spread over Eurasia, but independent origin in other regions cannot be ruled out [15,25,37]. 

## 4. Evolution of Some Genes (*Waxy*, *PPO*, *HD1*, *PRR37*, *SvLes1*, and *C*) under Human and/or Natural Selection

Several genes involved in domestication and diversification have been studied in cereals such as rice (e.g., [41,42]), maize (e.g., [43,44]), and six-rowed and naked grains in barley [45,46]. Recently, several genes involved in domestication, diversification, and adaptation have been identified in foxtail millet. Here, we review some of the genes involved in the evolution of foxtail millet, such as *Waxy* controlling amylose content in the endosperm, the *polyphenol oxidase* (*PPO*) gene for the phenol color reaction (Phr) in grains, two genes involved in heading date (*Heading date 1* [*HD1*] and *Pseudo-response regulator 37* [*PRR37*]), shattering genes (*qSh1*, *Sh1*, and *SvLes1*), and the *C* gene involved in leaf sheath pigmentation.

### 4.1. Waxy

The endosperm starch of cereals consists of amylose and amylopectin. Wild-type (non-waxy) endosperm starch consists of approximately 20% or more amylose and approximately 80% amylopectin, whereas the waxy (glutinous) type consists of approximately 100% amylopectin and lacks amylose. The non-waxy type controlled by *Wx* is genetically dominant compared with the waxy type controlled by *wx*. The texture of endosperm starch with the recessive genotype, the waxy type, is stickier than that of the normally dominant non-waxy type when cooked. Both endosperm types are found in landraces of sorghum, rice, foxtail millet, maize, common millet, barley, and Job’s tears [47]. Waxy types of these cereals are found in East and Southeast Asia but are rare in India and further west. A core area where people show a strong ethnobotanical preference for waxy cereals extends from Southern China through Northern Thailand and Laos to Northeastern India [47,48]. In adjacent areas, such as Taiwan, Japan, and Korea, waxy cereals are grown mainly on upland soils and are used in traditional rituals or eaten only on special occasions. This trait is associated with ethnological preferences in these areas (e.g., [49,50]). 

The waxy endosperm arises through the disrupted expression or loss of function of the *Waxy* (*GBSS1*) gene, which encodes granule-bound starch synthase I (GBSS I) [51]. Waxy-type cereals are characterized by little or no starch amylose, which constitutes approximately 20% or more of the total starch in the non-waxy endosperm. This food characteristic has frequently been neglected in other regions, although waxy maize, which was first reported in Chinese landraces [52], is now used globally to produce waxy corn starch. The molecular basis for artificial and spontaneous waxy mutants has also been elucidated [53]. Several mutations arise from the insertion of transposable elements into this gene. The molecular genetics of the *GBSS I* gene have also been studied in rice [54,55,56,57,58], barley [59,60,61], sorghum [62,63,64,65], Job’s tears [66], maize [67,68,69,70,71,72], and common millet [73]. In rice and barley, waxy landraces show a monophyletic origin [57,58], whereas the waxy landraces of sorghum [62,63,64,65], Job’s tears [66], maize [67,68,69,70,71,72], and common millet [73] originated polyphyletically. The origins of waxy in these cereals were reviewed by Fukunaga [74] and Gaur et al. [75].

Waxy phenotypes have also been observed in foxtail millets. The molecular basis of naturally occurring *wx* mutants in foxtail millet has been thoroughly examined [76,77,78,79]. Waxy foxtail millet probably evolved from the non-waxy type after domestication, as its wild ancestor had a non-waxy endosperm [76]. In addition to these two types, an intermediate- or low-amylose type of this crop has also been reported (Sakamoto 1987). Amylose content is positively correlated with the amounts of GBSS 1 protein in the three phenotypes [80] and is genetically controlled by *Waxy* (*GBSS 1*) alleles [76]. Other genes that regulate amylose content, such as *du* genes in rice [81], are unknown in *S. italica*. The sequence of full-length cDNA and the genomic structure of the *Waxy* (*GBSS 1)* gene in foxtail millet were determined, and a preliminary diversity analysis indicated multiple origins of waxy endosperm types [77]. Kawase et al. [78] analyzed 841 foxtail millet landraces and classified them into 11 types using PCR-based methods. They concluded that waxy foxtail millet had four independent origins, and low-amylose foxtail millet had three, according to insertions of transposable elements (Figure 2). Van et al. [79] found that the *waxy* gene of foxtail millet contains several SNPs and small indels. Hachiken et al. [82] also investigated sequence variations in this locus and revealed that the alleles of non-waxy accessions were more polymorphic than those of waxy and low-amylose accessions at the sequence level. This supports the hypothesis that the waxy and low-amylose types originated from the non-waxy type.

### 4.2. Variations in Phr and Evolution of the Polyphenol Oxidase (Si7PPO) Gene

Phr is a coloration of the hulls/lemma and palea (grains) of cereals after soaking in phenol solution, which is used for variety discrimination in rice [83] and barley [84]. Positive Phr types show black coloration after soaking in phenol solution, whereas negative Phr types do not show any coloration. Variations in Phr and the geographical distribution of Phr phenotypes in foxtail millet have been reported [85]. A respective study showed that Phr in foxtail millet is controlled by a single gene (positive Phr is dominant and negative Phr is recessive), and that the negative Phr type is predominantly distributed in Eurasia, whereas the positive Phr type generally has a skewed distribution toward subtropical and tropical regions, including Nansei Islands of Japan, Taiwan, the Philippines, Nepal, and India (21–100%). Inoue et al. [86] also investigated the underlying molecular mechanisms. The *polyphenol oxidase* (*Si7PPO*) gene responsible for Phr was isolated, and the molecular genetic basis of negative Phr that occurred in the crop evolution of foxtail millet was investigated. First, they found that a *PPO* gene homolog on chromosome 7 showed the highest similarity to *PPO* genes expressed in the hulls (grains) of other cereal species, including rice, wheat, and barley, and designated it *Si7PPO*. They also analyzed the genetic variation conferring a negative Phr reaction. Of 480 investigated accessions of the landraces, 87 (18.1%) showed a positive and 393 (81.9%) showed a negative Phr. In 393 Phr-negative accessions, three types of loss-of-function *Si7PPO* genes were predominant at various locations. One of them had an SNP in exon 1, resulting in a premature stop codon, and was designated “stop codon type”; a different type had an insertion of a transposon (*Si7PPO-TE1*) in intron 2 and was designated “TE1-insertion type”; and the other had a 6-bp duplication in exon 3, resulting in the duplication of two amino acids, and was designated “6-bp duplication type”. In addition, we identified several mutations in each of these three types. As a rare variant of the stop codon type, one accession additionally had an insertion of a transposon, *Si7PPO-TE2*, in intron 2 and was designated “stop codon + TE2 insertion type.” The geographical distribution of accessions with positive Phr and those with the three major types of negative Phr were also investigated (Figure 3). Accessions with positive Phr were found in subtropical and tropical regions at frequencies of approximately 25–67%, and those with negative Phr were broadly distributed in Europe and Asia. The stop codon type was found in 285 accessions and was broadly distributed in Europe and Asia, whereas the TE-1 insertion type was found in 99 accessions from Europe to Asia, but not in those from India. The 6-bp duplication type was found in only eight accessions from the Nansei Islands (Okinawa Prefecture) of Japan. They also analyzed Phr in the wild ancestor and concluded that the negative Phr type likely originated after the domestication of foxtail millet (Figure 3). Their study also suggested that the negative Phr of foxtail millet arose from multiple independent loss-of-function mutations of the *Si7PPO* gene, which proved advantageous under some environmental conditions and human selection, as also seen in rice [87] and barley [88]. 

Recently, Fukunaga et al. [89] carried out further analyses of the diversity and phylogeny of *Si7PPO*. The sequence polymorphism of the *Si7PPO* gene in 39 accessions consisting of foxtail millet landraces (32 accessions) and their wild ancestor ssp. *viridis* (seven accessions) collected from various regions in Europe and Asia was analyzed to elucidate the diversity and evolution of the *Si7PPO* gene. The accessions included the wild type (positive Phr) and three different types of loss-of-function phenotype (negative Phr), stop codon type, TE1-insertion type, and 6-bp duplication type examined in a previous study [86]. A phylogenetic tree of the gene was constructed, indicating that accessions with positive Phr showed higher genetic diversity at the nucleotide sequence level, and the three different loss-of-function types formed different clusters, strongly suggesting that landraces with negative Phr have multiple origins from three different lineages, including both landrace and ssp. *viridis* accessions with positive Phr; stop codon type originated from East Asian landraces with positive Phr; TE1-insertion type originated from Indian or Western European landraces with positive Phr; and landraces from Nansei Islands with a 6-bp duplication derived from Taiwanese landraces with positive Phr (Figure 3). Therefore, the variation in the *Si7PPO* locus is a robust indicator that helps trace crop evolution pathways after domestication in foxtail millet, as more than 80% of the landraces exhibit negative Phr.

### 4.3. Heading Time Genes Heading Date1 (HD1) and SiPRR37

Heading time is one of the most important traits for adapting to local environments. This trait has already been investigated in foxtail millet; landraces show high variability in heading time, and this trait is determined by a combination of the length of the basic vegetative growth period and sensitivity to short-day conditions [31,32]. A recent phylogenetic analysis showed that heading time was associated with the phylogenetic differentiation of foxtail millet landraces [90]. This trait is also known to vary in other plant species and has been investigated in detail [91,92]. Recently, the molecular mechanisms underlying this trait have been studied in several plant species, particularly in model plants such as rice and Arabidopsis [93].

The heading time of foxtail millet has been investigated in terms of genetic and quantitative-trait locus (QTL) analyses [94,95,96], and several candidate genes associated with heading time have been identified.

Two genes involved in heading time were investigated in detail: *SiHD1* and *SiPRR37*. *Heading date 1* (*HD1*) gene is a homolog of *CONSTANS (CO)* in Arabidopsis and of *HD1* in rice [93]. A splicing variant of this gene is common in European and Asian landraces of foxtail millet [97,98]. Fukunaga et al. [97] investigated the genetic variation in a rice *HD1* homolog in foxtail millet (i.e., *SiHD1*), and they found a nucleotide substitution at a putative splice site of intron 1 in *SiHD1* and proposed that accessions with a nucleotide substitution were carrying a splicing variant. They investigated the geographical distribution of the splicing variant in 480 accessions of foxtail millet from various regions of Europe, Asia, and Africa and 13 accessions of ssp. *viridis*, the wild ancestor; it was found that the splicing variant was broadly distributed in Europe and Asia (Figure 4) and that the wild type was predominant in the wild ancestor. The differences in heading time between accessions with the wild-type allele of the *SiHD1* gene and those with the spliced variant allele were unclear. Liu et al. [98] found the same variant in foxtail millet and concluded that this gene was involved in foxtail millet domestication, based on evidence of the strong selection (i.e., selective sweep) of this gene.

The other heading time-associated gene that was thoroughly examined is *Pseudo-response regulator 37* (*SiPRR37*). Recently, two research groups found that this gene plays an important role in the latitudinal adaptation of foxtail millet [99,100]. Li et al. [99] investigated Chinese landraces of foxtail millet using a genome-wide association study (GWAS) and found that transposable element (TE) insertion into *SiPRR37* was important for the adaptation of foxtail millet to Northeast China by genotyping 312 accessions (mostly from China), whereas Fukunaga et al. [100] found TE insertion in a Taiwanese landrace by QTL analysis of recombinant inbred lines derived from a hybrid between a Japanese landrace and a Taiwanese landrace. They found that TE insertions are predominantly distributed in Southern Asia, such as the Nansei Islands of Japan, Taiwan, the Philippines, Indonesia, Thailand, Myanmar, Bangladesh, India, Nepal, Pakistan, and Afghanistan, as well as in Kenya, and they sporadically occur in Ukraine and East Asia, including China, Korea, and Japan, according to the genotyping of 99 foxtail millet landraces from Eurasia and Africa (Figure 5). Both studies confirmed that this gene was key to latitudinal adaptation in foxtail millet, similar to other cereals, such as rice and sorghum [101,102].

### 4.4. Shattering Genes

The loss of seed shattering is one of the most important domestication-related traits in cultivated cereals, and the genetic basis of this trait has been thoroughly investigated in rice, maize, sorghum, and barley [41,103,104,105,106]. In foxtail millet, QTL mapping for seed shattering was performed, and QTLs for reduced shattering were found on chromosomes V (5) and IX (9) [107]. *Sh1* and *qSH1* are candidate genes for the QTLs on chromosomes IX and V, respectively. An 855-bp *PIF/Harbinger* MITE in exon 2 of *Sh1* reduces shattering. Furthermore, Liu et al. [108] investigated *Sh1* sequences in detail and performed a phylogenetic analysis of *Sh1*, suggesting a single origin of foxtail millet in China. Recently, a gene for the MYB transcription factor, *Less shattering* (*SvLes1*) on chromosome V, has also been identified as a QTL of seed shattering, according to a GWAS on *S. viridis* accessions [28]. It was found that *SvLes1* had two alleles in *S. viridis*, i.e., *SvLes1-1* and *SvLes1-2*. *SvLes1-1* is a wild-type allele showing high shattering, whereas *SvLes1-2* is a reduced-shattering allele with a nucleotide substation leading to an amino acid substitution from R to S at position 86. The reference sequence of strain A10.1 of *S. viridis* has the *SvLes1-2* allele, and 24% of the 215 accessions of the GWAS panel in *S. viridis* have this allele. In Yugu1, a Chinese cultivar from which the reference sequence of foxtail millet was sourced [26], this gene is disrupted by a *copia* TE (*copia38*) inserted into exon 2. This allele was designated *SiLes1-TE*. This result was confirmed using a CRISPR-Cas9 system to achieve the knockout of *SvLes1-1* [28]. The results strongly suggest that this gene is an important domestication gene in foxtail millet. TE insertions were found in this gene in 78 out of 79 accessions of foxtail millet [28]; however, no global respective study on foxtail millet landraces has been conducted so far. The assessment of TE insertions in the *SvLes1* on a global scale is essential to address the question of whether foxtail millet domestication has a monophyletic or polyphyletic history. Fukunaga et al. [109] screened 131 accessions of foxtail millet landraces and found that 16 landraces (12.2%) had no TE, despite showing non-shattering; they sequenced these 16 accessions and classified them into three alleles of this gene: *SvLes1-1*, *SvLes1-2*, and a new allele, *SvLes1-3*. The geographic distribution of these three alleles is different: *SvLes1-1* is distributed in Georgia, Germany, and Spain; *SvLes1-2* occurs in Afghanistan, Uzbekistan, and Belgium; and *SvLes1-3* is distributed in Japan, South Korea, and France (Figure 6). These results imply that the genetic basis of shattering in foxtail millet is more complex than previously assumed.

### 4.5. The C Gene Involved in Leaf Sheath Pigmentation

Variations in anthocyanin pigmentation are manifested in various parts of foxtail millet, such as the leaf sheaths, leaves, and bristles [90,110]. Plant pigmentation is a visible and easily distinguishable characteristic that may have been used to identify landraces during early agriculture. The results of GWASs suggested that anthocyanin pigmentation in leaves is regulated by multiple genes. The *C* gene, a gene for an MYB transcription factor [90], is one of the genes that control leaf sheath color. Recently, Fukunaga et al. [100] used mapping with double-digest restriction-site-associated sequencing to confirm the *C* gene as a responsible factor of leaf sheath pigmentation in recombinant lines inbred between a Japanese landrace with red leaf sheaths and a Taiwanese landrace with green leaf sheaths. They found an insertion of a transposable element into the *C* gene of the Taiwanese landrace used in mapping, resulting in loss of function of the gene, as well as an insertion of different TEs into the *C* gene of Yugu1, resulting in loss of function of this gene ([100]; Figure 7). The TE insertion in exon 3 was also reported by Liu et al. [111]. Recently, in addition to the *C* gene, *PPLS1* (purple color of the pulvinus and leaf sheath) on chromosome 7, which encodes a basic helix–loop–helix transcription factor, was confirmed to cause leaf sheath pigmentation [112]. Green leaf sheaths appear to have multiple origins. Fukunaga et al. are currently analyzing landraces with green leaf sheaths and the results will be presented elsewhere.

## 5. Perspective

Recent studies on phylogeny and association mapping using next-generation sequencing technology have further elucidated the relationships of foxtail millet and identified several candidate genes involved in the domestication and diversification of landraces [90,113]. Most phylogenetic studies indicated that foxtail millet had been domesticated in China, but a few studies implied that this millet had been domesticated in other regions such as Central and West Asia independently. Detailed phylogenetic studies using more accessions of its wild ancestor, ssp. *viridis*, from various regions of Eurasia could unravel the tangled issue. To date, genes involved in domestication traits, such as the non-shattering of grains [28,108,114], genes involved in latitudinal adaptation [98,100], and genes which evolved under human selection [77,78] have been investigated. Molecular studies on these genes indicated that some traits such as amylose content in endosperm, polyphenol oxidase in glume, and leaf sheath color had been selected multiple times, suggesting that this millet had evolved under preferences of the people and natural selection. Genetic mapping of a gene involved in bristle formation [115], panicle morphology [116] and QTLs for inflorescence structure [117], branching and height [114,118], and agronomic traits [119] has also been reported. Further analyses of the candidate genes will deepen our understanding of the domestication and crop evolution of foxtail millet.

Several researchers have proposed that TEs play an important role in the diversification of domesticated foxtail millet [21,77,78,86,100,108,111,113]. Consequently, foxtail millet represents an excellent model for the TE-mediated evolution of plant domestication and diversification.

## Figures and Tables

**Figure 1 plants-13-00218-f001:**
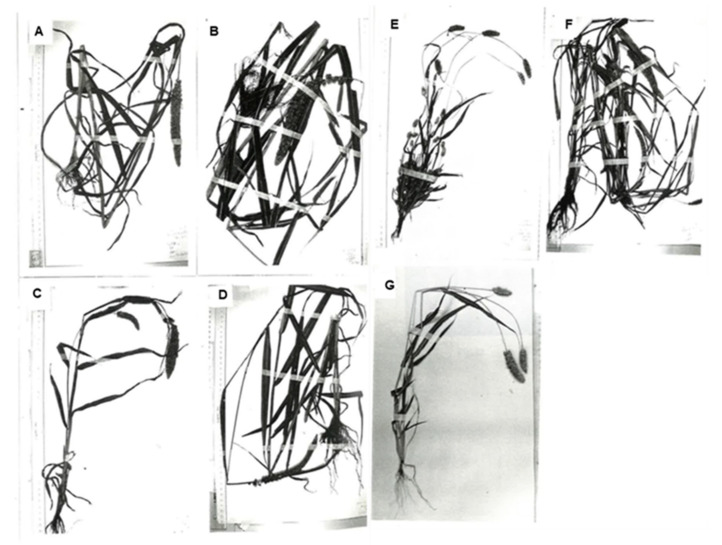
Specimens showing morphological variation among foxtail millet landraces [36] from (**A**) Japan; (**B**) Taiwan; (**C**) Northeast China; (**D**) the Philippines; (**E**) Saratov, Russia; (**F**) India; (**G**) Bulgaria.

**Figure 2 plants-13-00218-f002:**
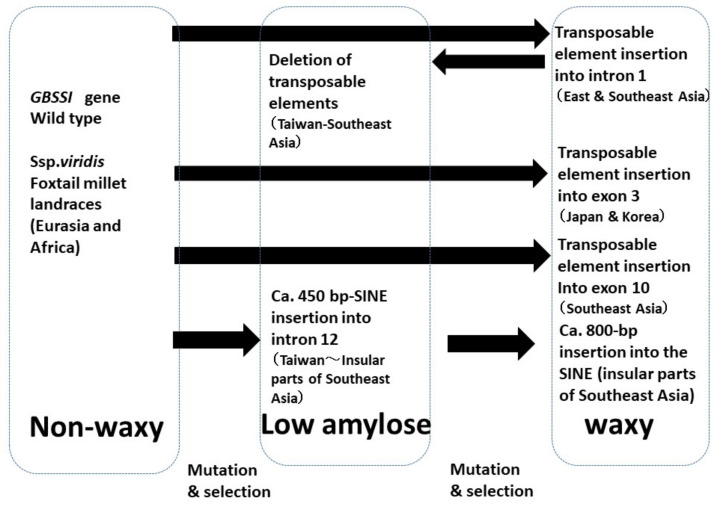
A schematic illustration of the evolution of waxy and low-amylose types of foxtail millet according to Kawase et al. [78].

**Figure 3 plants-13-00218-f003:**
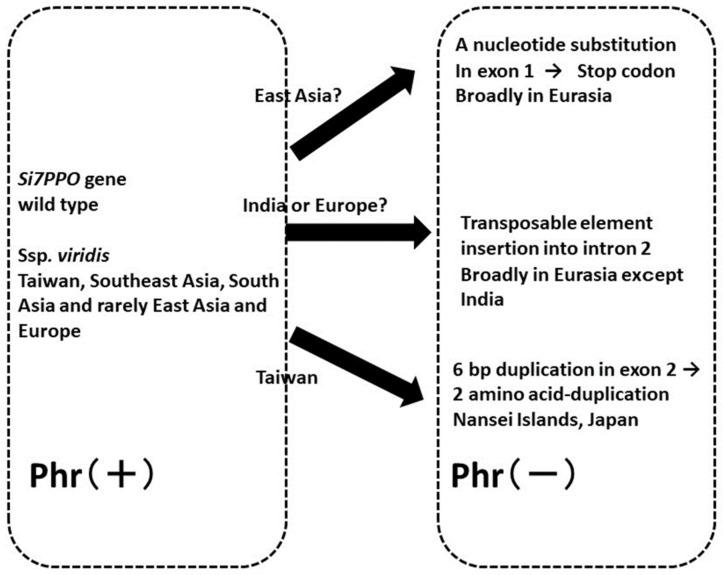
A schematic illustration of the evolution from positive Phr (wild type) to three main genotypes of Phr, stop codon type, TE1 insertion type, 6-bp duplication type, and stop codon according to Inoue et al. [86] and Fukunaga et al. [89].

**Figure 4 plants-13-00218-f004:**
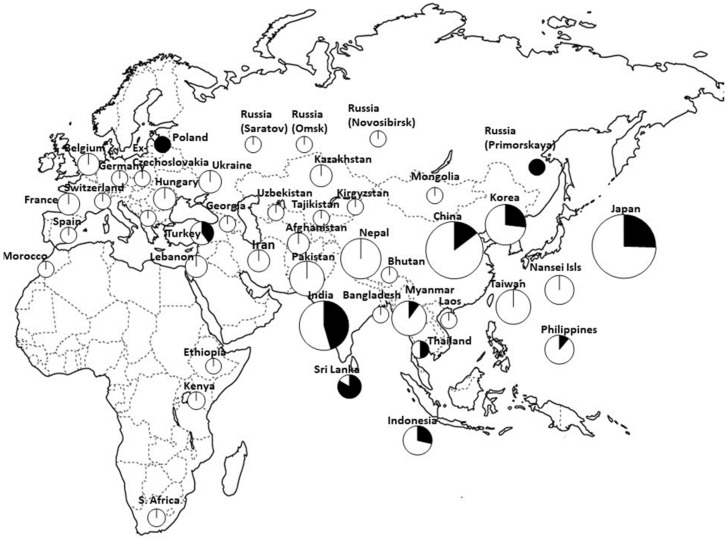
Geographical distribution of the *HD1* gene, wild type, and splicing variant [97].

**Figure 5 plants-13-00218-f005:**
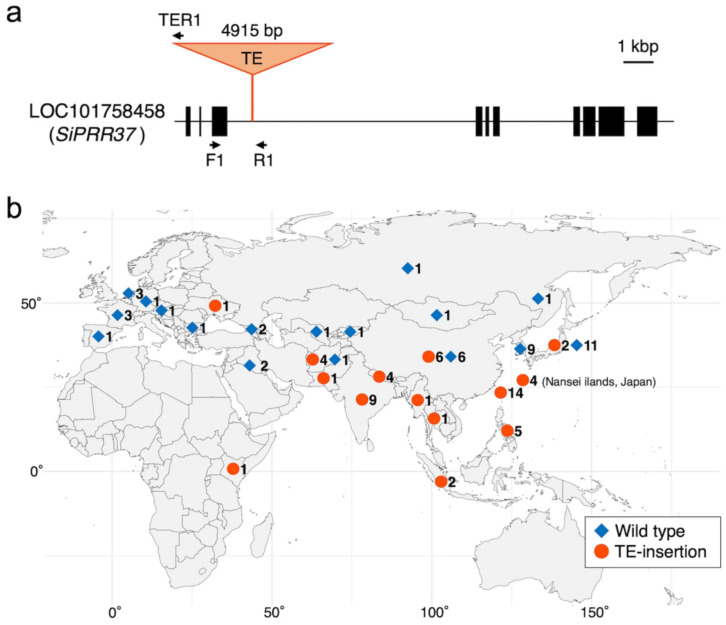
The variant of *SiPRR37* with TE-insertion and its geographical distribution. (**a**) Structure of the *SiPRR37* gene and insertion of transposable element (TE). Primers used for the analysis of TE-insertion were shown as arrows. (**b**) Geographical distribution of TE-insertion type and non-insertion type (Wild type) of *SiPRR37* gene of foxtail millet. The number next to the symbol indicates the number of accessions checked in this study. The map was created using the R package “maps” ver. 3.3.0 (https://CRAN.R-project.org/package=maps). The X-axis and the Y-axis indicate longitude and latitude, respectively [100].

**Figure 6 plants-13-00218-f006:**
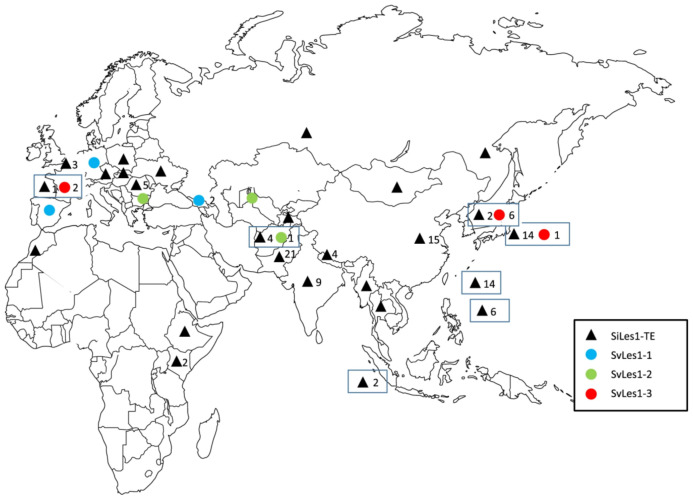
Geographical distribution of shattering gene alleles, *SvLes1* alleles, *SvLes1-TE*, *SvLes1-1*, *SvLes1-2*, and *SvLes1-3* in foxtail millet [109]. Black triangles, blue dots, green dots and red dots indicate *SvLes1-TE*, *SvLes1-1*, *SvLes1-2* and *SvLes1-3*, respectively. The number next to the symbol indicates the number of accessions checked in this study.

**Figure 7 plants-13-00218-f007:**
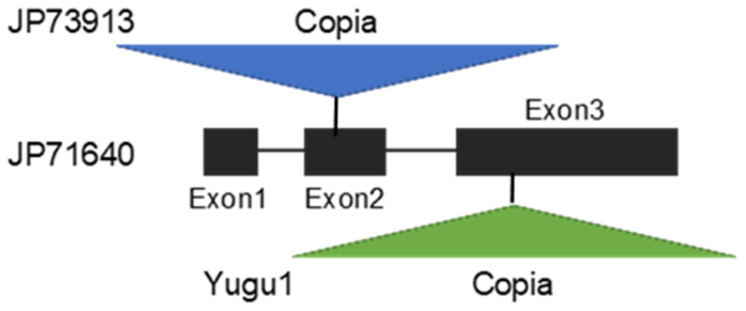
Structure of the *C* gene involved in anthocyanin pigmentation in foxtail millet and different TE insertions in the gene causing green leaf sheaths [100].

**Table 1 plants-13-00218-t001:** Genetic works on genetic differentiation of foxtail millet landraces and geographical groups and center of diversity revealed by the studies (Fukunaga [3] modified).

Genetic Markers/Intraspecific Hybrid Pollen Sterility	Geographical Groups	Center of Diversity	References
Esterase isozymes	East Asia vs. Europe	East Asia	[39]
10 isozymes	China–Korea–Japan, Okinawa (Nansei Islands of Japan)–Taiwan, India–Kenya, Europe		[9]
Prolamine	Europe, Tropical Groups	China	[13]
Hybrid sterility	China–Korea–Japan, Okinawa (Nansei Islands of Japan)–Taiwan, Lan–Hsű–Batan Islands, India–Afghanistan, Europe		[25]
rDNA	China–Korea–Japan, Okinawa (Nansei Islands of Japan)–Taiwan–the Philippines, India, Afghanistan–Northern Pakistan	China	[14,15,17]
Nuclear RFLP	East Asia, Nansei Islands–Taiwan–the Philippines, India, Afghanistan–Central Asia–Europe	China	[21]
mtDNA	Not clear	China	[22]
RAPD	Central Europe and two Asiatic groups (north and south)		[19]
AFLP	Not clear	China	[20]
TD	East Asia, Nansei Islands–Taiwan–the Philippines, India, Central Asia, Europe	China	[23]
GBS	Four clusters 1. Southern Asia (Nepal, Pakistan, Afghanistan, Iran, Turkey, and the Near East) + Western and Northern Europe, 2. Far East focus (including Japan, North and South Korea, and the region of Russia bordering the Sea of Japan) + Western China + sporadically Eastern Europe, 3. India, Sri Lanka, Bangladesh, and Eastern and Southern Africa, 4. widespread east–west distribution.		[40]

## Data Availability

The raw data supporting the conclusions of this article will be made available by the authors on request.

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
