# Peer review of "Crop Evolution of Foxtail Millet"

_plants, 2024, doi:10.3390/plants13020218_

Round 1
Reviewer 1 Report
Comments and Suggestions for Authors
This manuscript provides a comprehensive review on the domestication, genetic differentiation, and evolution of foxtail millet. It summarizes key findings from foxtail millet research using morphological characterization, biochemical markers, DNA markers, and analysis of important genes involved in domestication traits. The review covers the newest knowledge on the origin and spread of foxtail millet, as well as the genetic basis underlying important agronomic traits. The manuscript would serve as a valuable reference for foxtail millet researchers.
Questions to Authors
(1) The very wide scope of the review means that the coverage of some topics is quite brief. More details on key studies regarding the origin, spread, and differentiation of foxtail millet could enrich the review.
(2) Could you expand on the evidence regarding independent domestications of foxtail millet outside of China? Which regions show primitive landraces that may represent local domestications?
(3) For the genes reviewed, are there clear patterns in terms of which traits show single vs multiple evolutionary origins? What implications does this have for foxtail millet evolution under selection?
(4) How do you see genomic studies and association mapping enhancing our understanding of foxtail millet domestication and differentiation in future? What key gaps need to be addressed?
Author Response
Thank you for your comments. They are helpful to enrich our manuscript.
1. The very wide scope of the review means that the coverage of some topics is quite brief. More details on key studies regarding the origin, spread, and differentiation of foxtail millet could enrich the review.
Thank you for your comments. We added some interpretation of the phylogenetic data. E.g., we added a sentence “These results indicate that this millet was domesticated in China and spread over Eurasia but independent origin in other regions cannot be ruled out.” We also added similar sentence at the end of p.3 and Perspective.
2. Could you expand on the evidence regarding independent domestications of foxtail millet outside of China? Which regions show primitive landraces that may represent local domestications?
I added sentences “Most phylogenetic studies indicated that foxtail millet had been domesticated in China but a few studies implied that this millet had been domesticated in other regions such as Central and West Asia independently. Detailed phylogenetic studies using more accessions of its wild ancestor, ssp. viridis from various regions of Eurasia could unravel the tangled issue” in Perspective.
3.For the genes reviewed, are there clear patterns in terms of which traits show single vs multiple evolutionary origins? What implications does this have for foxtail millet evolution under selection?
Thank you for your comments. We added sentences “ Molecular studies on these genes indicated that some traits such as amylose content in endosperm, polyphenol oxidase in glume and leaf sheath color had been selected multiple times, suggested that this millet had evolved under preferences of the people and natural selection” in Perspective.
(4) How do you see genomic studies and association mapping enhancing our understanding of foxtail millet domestication and differentiation in future? What key gaps need to be addressed?
Thank you for the comments. As I have already written,
“Genetic mapping of a gene involved in bristle formation [114], panicle morphology [115] and QTLs for inflorescence structure [116], branching and height [117,118], and agronomic traits [119] have also been reported. Further analyses of the candidate genes will deepen the understanding of the domestication and crop evolution of foxtail millet”.
Reviewer 2 Report
Comments and Suggestions for Authors
This review manuscript is expected to provide valuable insights into the genetic differentiation of foxtail millet landraces collected from various parts of Europe, Asia, and Africa. The investigation encompasses morphological/agronomic characteristics, biochemical markers, intraspecific hybrid pollen sterility, and DNA markers. Additionally, recent studies on genes associated with the domestication and diversification of foxtail millet are discussed in this review. The focus is on Waxy, conferring amylose content in the endosperm; Si7PPO, controlling polyphenol oxidase; HD1 and SiPRR37, controlling heading time; Sh1 gene and SvLes1, which are involved in grain shattering; and the C gene, controlling leaf sheath pigmentation.
Considering the insightful implications of this review manuscript, I recommend that it be considered for publication in the journal "Plants."
Author Response
This review manuscript is expected to provide valuable insights into the genetic differentiation of foxtail millet landraces collected from various parts of Europe, Asia, and Africa. The investigation encompasses morphological/agronomic characteristics, biochemical markers, intraspecific hybrid pollen sterility, and DNA markers. Additionally, recent studies on genes associated with the domestication and diversification of foxtail millet are discussed in this review. The focus is on Waxy, conferring amylose content in the endosperm; Si7PPO, controlling polyphenol oxidase; HD1 and SiPRR37, controlling heading time; Sh1 gene and SvLes1, which are involved in grain shattering; and the C gene, controlling leaf sheath pigmentation.
Considering the insightful implications of this review manuscript, I recommend that it be considered for publication in the journal "Plants."
Reply:
Thank you for your review and comments. I hope that this review will help several researchers who are interested in crop evolution and genetic resources of crop evolution.
Reviewer 3 Report
Comments and Suggestions for Authors
Dear sir,
the present article 'Crop Evolution of Foxtail Millet' is clearly written and the message is understandable. Authors provide multiple evidence on foxtail millet domestication. But they don´t conclude or they are not sure about the domestication events on this crop based on their conclusions. It is one of the few articles I have reviewed where the document supply enough evidence (and well written), but they fail to provide any solid conclusion/s. I would suggest authors to write at least three conclusions from their study, e.g., place and time of first domestication, wild ancestor, secondary events of domestication, etc. Even if they are not secure 100%, at least it is likely that evidence restricts domestication to certain places and time. I am uploading an annotated file of the document, with minor corrections.
Abstract. Authors refers to China as the main center of domestication, but this is not their conclusion. I think your guess is foxtail millet might be domesticated in Central Asia, isn´t it?
Conclusions. The results are powerful but the conclusions and the own abstract are rather weak. It seems you don´t have a clear conclusion, and the understanding of domestication is yet to be seen.
Best

Comments on the Quality of English LanguageDear sir,
the present article 'Crop Evolution of Foxtail Millet' is clearly written and the message is understandable. Authors provide multiple evidence on foxtail millet domestication. But they don´t conclude or they are not sure about the domestication events on this crop based on their conclusions. It is one of the few articles I have reviewed where the document supply enough evidence (and well written), but they fail to provide any solid conclusion/s. I would suggest authors to write at least three conclusions from their study, e.g., place and time of first domestication, wild ancestor, secondary events of domestication, etc. Even if they are not secure 100%, at least it is likely that evidence restricts domestication to certain places and time. I am uploading an annotated file of the document, with minor corrections.
Abstract. Authors refers to China as the main center of domestication, but this is not their conclusion. I think your guess is foxtail millet might be domesticated in Central Asia, isn´t it?
Conclusions. The results are powerful but the conclusions and the own abstract are rather weak. It seems you don´t have a clear conclusion, and the understanding of domestication is yet to be seen.
Best
Author Response
the present article 'Crop Evolution of Foxtail Millet' is clearly written and the message is understandable. Authors provide multiple evidence on foxtail millet domestication. But they don´t conclude or they are not sure about the domestication events on this crop based on their conclusions. It is one of the few articles I have reviewed where the document supply enough evidence (and well written), but they fail to provide any solid conclusion/s. I would suggest authors to write at least three conclusions from their study, e.g., place and time of first domestication, wild ancestor, secondary events of domestication, etc. Even if they are not secure 100%, at least it is likely that evidence restricts domestication to certain places and time. I am uploading an annotated file of the document, with minor corrections.
Abstract. Authors refers to China as the main center of domestication, but this is not their conclusion. I think your guess is foxtail millet might be domesticated in Central Asia, isn´t it?
Conclusions. The results are powerful but the conclusions and the own abstract are rather weak. It seems you don´t have a clear conclusion, and the understanding of domestication is yet to be seen.
Best
Reply: Thank you for the comments and several minor corrections. They are very helpful.
We added a sentence “These results indicate that this millet was domesticated in China and spread over Eurasia but independent origin in other regions cannot be ruled out”. We also added some sentences on origin of foxtail millet in Perspective.